# Genetic and Molecular Mechanisms in Brugada Syndrome

**DOI:** 10.3390/cells12131791

**Published:** 2023-07-05

**Authors:** Errol Moras, Kruti Gandhi, Bharat Narasimhan, Ramon Brugada, Josep Brugada, Pedro Brugada, Chayakrit Krittanawong

**Affiliations:** 1Department of Medicine, Icahn School of Medicine at Mount Sinai, New York, NY 10029, USA; 2Debakey Cardiovascular Institute, Houston Methodist Hospital, Houston, TX 77030, USA; 3Cardiology, Cardiac Genetics Clinical Unit, Hospital Universitari Josep Trueta, Hospital Santa Caterina, 17007 Girona, Spain; 4Cardiovascular Genetics Center and Clinical Diagnostic Laboratory, Institut d’Investigació Biomèdica Girona-IdIBGi, 17190 Salt, Spain; 5Cardiovascular Institute, Hospital Clínic, 08036 Barcelona, Spain; 6Pediatric Arrhythmia Unit, Hospital Sant Joan de Déu, 08950 Barcelona, Spain; 7Department of Medicine, University of Barcelona, 08036 Barcelona, Spain; 8Cardiovascular Division, Free University of Brussels (UZ Brussel) VUB, B-1050 Brussels, Belgium; 9Medical Centre Prof. Brugada, B-9300 Aalst, Belgium; 10Arrhythmia Unit, Helicopteros Sanitarios Hospital (HSH), Puerto Banús, 29603 Marbella, Spain; 11Cardiology Division, NYU Langone Health and NYU School of Medicine, New York, NY 10016, USA

**Keywords:** genetic, genes, Brugada syndrome, Brugada

## Abstract

Brugada syndrome is a rare hereditary arrhythmia disorder characterized by a distinctive electrocardiogram pattern and an elevated risk of ventricular arrhythmias and sudden cardiac death in young adults. Despite recent advances, it remains a complex condition, encompassing mechanisms, genetics, diagnosis, arrhythmia risk stratification, and management. The underlying electrophysiological mechanism of Brugada syndrome requires further investigation, with current theories focusing on abnormalities in repolarization, depolarization, and current-load match. The genetic basis of the syndrome is strong, with mutations found in genes encoding subunits of cardiac sodium, potassium, and calcium channels, as well as genes involved in channel trafficking and regulation. While the initial discovery of mutations in the SCN5A gene provided valuable insights, Brugada syndrome is now recognized as a multifactorial disease influenced by several loci and environmental factors, challenging the traditional autosomal dominant inheritance model. This comprehensive review aims to provide a current understanding of Brugada syndrome, focusing on its pathophysiology, genetic mechanisms, and novel models of risk stratification. Advancements in these areas hold the potential to facilitate earlier diagnosis, improve risk assessments, and enable more targeted therapeutic interventions.

## 1. Introduction

Brugada syndrome (BrS) is an autosomal dominant, inherited cardiac channelopathy associated with an elevated risk of sudden cardiac death (SCD) in young adults. Following the early descriptions by Osher and Wolff in 1953 as normal variants, it was later characterized by Pedro and Josep Brugada in 1992 as associated with SCD [1]. The global prevalence of BrS is estimated at around 0.05%, while the more common Brugada pattern is seen in around 0.4% of individuals [2,3]. Among young adults without structural heart disease, BrS is implicated in up to 20% of SCD, while this association is weaker in infants and children [4,5,6]. The highest prevalence of BrS is reported in Southeast Asia—up to 14 times higher than the global prevalence [7,8]. In this region, it is recognized as the leading cause of natural death in males under the age of 50. This higher prevalence may be associated with specific sequences in the promoter region of the SCN5A gene [9,10,11,12,13]. To date, more than 300 different mutations in 19 specific genes have been identified as potentially contributive to the development of BrS [14]. However, these mutations only cumulatively account for 30–35% of diagnosed cases. This review aims to provide a comprehensive overview of our current understanding of this challenging condition with a specific focus on the pathophysiology as well as genetic and molecular mechanisms. Advances in this area will undoubtedly help facilitate earlier diagnosis and improved risk stratification as well as potentially more targeted therapeutics.

## 2. Pathophysiology

The current understanding of the pathophysiology of BrS has evolved significantly in recent years. The widely supported mechanism for BrS suggests that it is primarily caused by a repolarization disorder, characterized by abnormal shortening of the epicardial action potential duration [15,16,17,18,19]. However, an alternative depolarization disorder hypothesis, which focuses on conduction slowing, has also been proposed, based on clinical and experimental studies [19,20,21]. In this section, we provide a comprehensive and updated overview of the proposed pathophysiological mechanisms of BrS, including the repolarization and depolarization hypotheses, structural derangements, and the presence of node-like tissues (Figure 1).

### 2.1. Cellular and Ionic Mechanisms

In general, the cardiac action potential (AP) is a result of intricate interactions among ion channels, membrane voltages, ionic environments, and regulatory molecules. The rate at which the AP repolarizes is a crucial characteristic of cardiac cells, and alterations caused by disease or drugs can result in life-threatening arrhythmias. The initiation of the AP is primarily controlled by a specific type of fast sodium ion channel, which produces a large and rapid inward current (I_Na_) to depolarize the membrane during the initial upstroke of the AP. On the other hand, the repolarization phase of the AP is a slower process, influenced by a delicate balance between various types of ion channels that carry depolarizing membrane currents inward and repolarizing membrane currents outward. During the AP plateau, the primary depolarizing current is I_Ca(L),_ which decreases progressively due to voltage- and calcium-dependent inactivation mechanisms. To gradually shift the balance toward outward repolarization, I_Kr_ and I_Ks_ currents increase during the plateau, peaking at the late plateau phase. These diverse mechanisms align with the electrophysiological function of the cell [22].

Evidence suggests that BrS is associated with the amplification of intrinsic heterogeneities of the AP between epicardial and endocardial myocytes. The debate regarding the predominant electrophysiological mechanism of BrS has largely revolved around the repolarization versus depolarization hypotheses. Early studies on right ventricle (RV) wedge preparation by Potocskai et al. put forward the idea that late fractionated low-voltage potentials observed in the RV epicardium are associated with repolarization abnormalities [23]. They further suggested that radiofrequency ablation (RFA) of abnormal epicardial sites can eliminate these repolarization abnormalities, resulting in the normalization of the BrS electrocardiogram (ECG) pattern and a reduction in ventricular arrhythmias. In a study involving 135 symptomatic BrS patients, Pappone et al. demonstrated that localization and the elimination of abnormal electrical activity in the epicardial right ventricle outflow tract (RVOT) on ajmaline administration helped identify extensive areas of arrhythmogenic electrophysiological substrates (AES), which were successfully eliminated by RFA, leading to ECG normalization and the non-inducibility of ventricular tachycardia/ventricular fibrillation (VT/VF). This substrate-based ablation approach has shown promise in treating the arrhythmic consequences of BrS [24]. During phases 1 and 2 of the epicardial AP, an outward shift of ionic currents occurs, either via a reduction in the inward current (I_Na_ or I_Ca_) or an increase in the outward currents (I_kr_ or I_K-ATP_), which allows for the already dominant I_to_ to accentuate phase 1 depolarization. This accentuation occurs when phase 1 repolarizes to a voltage level below the range required to activate L-type Ca channels, leading to a failure of activation. As a result, a loss of the AP plateau, particularly in the RV subepicardial cells (where I_to_ is most prominent) occurs, leading to the development of an electrically vulnerable substrate. This increases the likelihood of phase 2 re-entry, leading to more frequent extrasystoles during this vulnerable period, which in turn can trigger ventricular arrhythmias [4]. During slow cardiac rates, disruptions of rapid inactivation produce a persistent sodium current, leading to an extended repolarization period. Meanwhile, an increase in the intermediate kinetic component of slow inactivation delays the restoration of sodium channels, resulting in decreased sodium currents and shortened AP durations at high heart rates [25]. The spread of the action potential dome from epicardial sites to those lacking it may underlie the R-on-T phenomenon while the rapid and inverse repolarization gradients contribute to ECG ST-segment elevation and T-wave inversion [16]. Electrogram readings of the RVOT in BrS patients have also revealed a combination of high repolarization gradients and delayed repolarization, potentially leading to VT and VF due to imbalances in calcium and potassium currents. Broadly, this mechanism forms the basis of the repolarization hypothesis [26].

On the other hand, the depolarization hypothesis suggests that abnormalities in the initial phases of depolarization lead to slow conduction in the RVOT, which underpins the clinical and electrocardiographic manifestations of BrS [26]. This is supported by the observations of Nademanee et al., who demonstrated the presence of late potentials and fractionated electrograms recorded from the RVOT using bipolar electrograms [27]. They reported that the RFA of these epicardial sites significantly reduces vulnerability to arrhythmias and the ECG manifestations of BrS. Genetic variations in the SCN5A gene, affecting the inward current during phase 0, are implicated in approximately a quarter of Brugada syndrome (BrS) cases, leading to a slower upstroke and delayed action potential formation that significantly contributes to ventricular arrhythmogenesis [28]. Moreover, fibrotic lesions observed in the RV may account for the presence of right bundle branch block and ST-segment elevation on electrocardiograms [29]. Multiple studies investigating the impact of SCN5A disruption in animal models have shown that mice with targeted SCN5A mutations exhibit decreased conduction velocity and the development of interstitial fibrosis [30,31,32]. Recent human studies using panoramic ventricular mapping have also demonstrated electrogram lengthening and fragmentation, indicating a decrease in conduction velocity and an increase in scattering [33].

An alternative viewpoint may be a heterogeneous disease with two components: a clinical entity linked to epicardial fibrosis causing conduction abnormalities and an independent pure ion channelopathy responsible for repolarization defects [34]. A study by Hoogendijk et al. puts forth another hypothesis that ventricular arrhythmias in BrS could be attributed to a subepicardial phenomenon known as the current-to-load mismatch. They suggested that a reduction in the sodium current, resulting from a channel malfunction or pore size, could lead to subepicardial excitation failure or delayed activation, while computational model simulations indicated that an imbalance between inward and outward currents might disrupt excitation and contribute to ST-segment elevation observed in BrS [35,36].

In summary, human studies offer the strongest support for the repolarization theory in BrS. Ischemia-based models derived from depolarization theories do not directly apply to the condition, while arrhythmias and channel mutations in I_Na_, I_to_, and I_CaL_ align with the repolarization theory. Excitation failure due to the current-to-load mismatch, regulated by specific ion currents, may explain the observed ST-segment elevation in Brugada patients. However, further human studies are necessary to obtain definitive data and resolve the existing controversy with more certainty.

### 2.2. Genetic Mechanisms

Brugada syndrome (BrS) is inherited through an autosomal dominant pattern of transmission. Researchers have identified mutations in 19 different genes, though only SCN5A is considered a definitive gene associated with the disease, while the rest of the genes are considered to have a weaker association (Table 1). These genes disrupt the normal function of ion channels involved in cardiac electrical activity, leading to a decrease in inward sodium/calcium currents or an increase in outward potassium currents. This disrupts the ionic balance of currents during the early stages of the action potential, resulting in the pro-arrhythmia of BrS [37].

### 2.3. Mutations Causing a Loss of Function of Sodium Channel Currents

The sodium channel is a voltage-gated, complex structure consisting of multiple subunits. The alpha subunit forms the aqueous pore responsible for channel opening, ion selectivity, and rapid inactivation. Additionally, smaller beta subunits play a role in regulating voltage sensitivity, gating kinetics, channel density, and localization on the cell surface [38,39]. Currently, nine alpha subunits (Nav1.1–Nav1.9) have been thoroughly studied and characterized. Among them, Nav1.5 is the primary sodium channel expressed in cardiac myocytes and plays a critical role in controlling their excitability [40,41]. The sodium channel is a transmembrane protein of considerable size that consists of four internally similar domains (DI–DIV), each with six segments that span the membrane (S1–S6). Among them, the S4 segment acts as a crucial voltage sensor due to its abundant positive residues. The extracellular loops of the S5 and S6 segments, known as P-loops, determine the ion selectivity of the channel [42]. While other voltage-gated sodium channels are present in the heart, they are unable to compensate for any dysfunction in NaV1.5. This highlights the pivotal importance of NaV1.5 in cardiac physiology [38].

**Table 1 cells-12-01791-t001:** Mutations associated with the Brugada syndrome.

Gene	Protein	Current	Effect	Function
** * SCN5A * **	** Na_V_1.5 (α-subunit of the voltage-dependent Na^+^ channel) **	**↓ *I_Na_***	**loss of function**	** mediator of the depolarizing inward sodium *I_Na_* current **
** *SCN1B* **	**β1-subunit of the voltage-dependent Na^+^ channel**	**↓ *I_Na_***	**loss of function**	**auxiliary protein modulator of Na_V_1.5 and the *I_Na_* current**
** *SCN2B* **	**β2-subunit of the voltage-dependent Na^+^ channel**	**↓ *I_Na_***	**loss of function**	**auxiliary protein modulator of Na_V_1.5 and the *I_Na_* current**
** *SCN3B* **	**β3-subunit of the voltage-dependent Na^+^ channel**	**↓ *I_Na_***	**loss of function**	**auxiliary protein modulator of Na_V_1.5 and the *I_Na_* current**
** *SCN10A* **	**Na_V_1.8 (α-subunit of the neuronal voltage-dependent Na^+^ channel)**	**↓ *I_Na_***	**loss of function**	**mediator of the depolarizing phase of the neural AP, associated with pain perception**
** *CACNA1C* **	**Ca_V_1.2 (α1C-subunit of the voltage-dependent L-type Ca^2+^ channel)**	**↓ *I_CaL_***	**loss of function and combined phenotype of BrS and SQTS**	**mediator of the inward calcium *I_CaL_* current**
** *CACNB2b* **	**β2-subunit of the voltage-dependent L-type Ca^2+^ channel**	**↓ *I_CaL_***	**loss of function and combined phenotype of BrS and SQTS**	**auxiliary protein modulator of Ca_V_1.2 and the *I_CaL_* current**
** *KCND3* **	**K_V_4.3 (α-subunit of the voltage-dependent K^+^ channel)**	**↑ *I_to_***	**gain of function**	**mediator of the transient outward K^+^ *I_to_* current**
** *KCNE3* **	**minK-related peptide 2 (β-subunit of the voltage-dependent K^+^ channel)**	**↑ *I_to_***	**gain of function**	**regulator of K_V_4.3**
** *KCNAB2* **	**β2-subunit of the voltage-dependent K^+^ channel**	**↑ *I_to_***	**gain of function**	**interaction with K_V_4.3**
** *KCND2* **	**K_V_4.2 (voltage-dependent K^+^ channel)**	**↑ *I_to_***	**gain of function**	**contributor to the transient outward K^+^ *I_to_* current**
** *KCNE5* **	**minK-related peptide 4 (β-subunit of the voltage-dependent K^+^ channel)**	**↑ *I_to_***	**gain of function**	**inhibitor of the delayed rectifying K_V_7.1 channel and modulator of K_V_4.3**
** *KCNJ8* **	**Kir6.1 (inward-rectifier K^+^ channel and subunit of the ATP-sensitive K^+^ channel)**	**↑ *I_K-ATP_***	**Gain of function**	**mediator of the *I_K-ATP_* currents**
** *ABCC9* **	**SUR2 (sulfonylurea receptor and subunit of the ATP-sensitive K^+^ channel)**	**↑ *I_K-ATP_***	**gain of function**	**modulator of *I_K-ATP_* currents**
** *KCNH2* **	**K_V_11.1/hERG (α-subunit of the voltage-dependent K^+^ channel)**	**↑ *I_Kr_***	**gain of function**	**mediator of the rapid component of the delayed rectifying potassium *I_Kr_* current**
** *CACNA2D1* **	**α2/δ subunit of the voltage-dependent L-type Ca^2+^ channel**	**↓ *I_CaL_?***	**Loss of function? and combined phenotype of SQTS and BrS**	**auxiliary protein modulator of Ca_V_1.2 and the *I_CaL_* current**
** *HCN4* **	**hyperpolarization-activated and cyclic nucleotide-gated ion channel 4**	**↓ *I_f_?***	**loss of function?**	**mediator of the pacemaker current, *I_f_***
** *TRPM4* **	**transient receptor potential melastatin 4**		**loss of function/gain of function**	**regulator of conduction and cellular electrical activity, which impact heart development**
** *FGF12* **	**fibroblast growth factor 12**	**↓ *I_Na_***	**interaction with Na_V_1.5 trafficking**	**modulator of Nav1.5 and the *I_Na_* current**
** *GPD1L* **	**glycerol-3-phosphate dehydrogenase 1-like**	**↓ *I_Na_***	**interaction with Na_V_1.5 trafficking**	**modulator of Na1.5 and the *I_Na_* current**
** *SLMAP* **	**Sarcolemma-associated protein (striatin-interacting phosphatase and kinase complex)**	**↓ *I_Na_***	**interaction with Na_V_1.5 trafficking**	**present in the T-tubules and regulator of excitation–contraction coupling**
** *PKP2* **	**plakophillin-2**	**↓ *I_Na_***	**changes in Na_V_1.5 expression in intercalated disc**	**binds to and modulates Na_V_1.5 and the *I_Na_* current**
** *SEMA3A* **	**semaphorin-3A**	**↑ *I_to_***	**loss of function**	**inhibitor of the K_V_4.3 channel**
** *RANGRF* **	**MOG1 (multicopy suppressor of Gsp1)**	**↓ *I_Na_*?**	**interaction with Na_V_1.5 trafficking**	**involved in nuclear protein import and regulates cell surface location of Na_V_1.5**
** *HEY2* **	**CHF1 (cardiovascular helix–loop–helix factor 1)**	**↑ *I_to_*?**	**Interaction with KCNIP2**	**transcriptional regulator of cardiac electrical function**

**↓** loss of function; **↑** gain of function.

**SCN5A:** This gene was first identified in 1995 with the help of fluorescence in situ hybridization [43]. It is located on chromosome 3P 21 and encodes the alpha subunit of Nav1.5, the importance of which has been outlined in the prior section. This gene was first linked to BrS in 1998 by Chen et al. and is responsible for producing the α-subunit of the voltage-gated cardiac sodium channel (Na v 1.5) [44]. Over 300 mutations related to SCN5A and associated with BrS have been identified in recent years, accounting for the majority of genetically confirmed cases [45]. Loss-of-function variants in SCN5A are commonly associated with Brugada syndrome (BrS). These variants are typically found in specific regions, such as the area between the first and second domains (DI and DII), the intracellular connection between the third and fourth domains (DIII and DIV), the P ring, and the D-terminal region of DIII. Examples of SCN5A polymorphisms linked to BrS include H558R and R34C [45,46,47,48,49]. However, these mutations are identified in only 11% to 28% of all individuals diagnosed with BrS [50]. The penetrance of SCN5A mutations has been found to exhibit incomplete penetrance and variable expression in pedigrees with Brugada syndrome, suggesting a complex inheritance pattern wherein other genetic variants may influence the phenotype. Genotype-negative individuals from SCN5A-positive pedigrees have also shown the type 1 Brugada ECG pattern, further indicating the Influence of additional genetic factors [51]. SCN5A mutations are not limited to BrS but are associated with various other cardiac conditions, which include long QT syndrome (LQT3), cardiac conduction system dysfunction, myocardial contractile dysfunction, dilated cardiomyopathy, heart failure, sick sinus syndrome, familial atrial standstill, atrial fibrillation, ventricular arrhythmias (such as long QT syndrome type 3 and idiopathic ventricular fibrillation), and progressive cardiac conduction defect (Lenegre syndrome). A substantial majority of these variants are non-synonymous, with a smaller portion resulting from deletion and duplication [42].

**GPD1-L:** Over a decade ago, Weiss et al. discovered a new locus associated with BrS located adjacent to the SCN5A gene [52]. This was later identified as the GPD1-L gene, which encodes the glycerol-3-phosphate dehydrogenase 1-like protein. The GPD1-L protein is closely linked, structurally and functionally, to the Na v 1.5 sodium channel [53]. The enzymatic dysfunction of GPD1-L ultimately leads to a reduction in iNa through the GPD1-L-dependent phosphorylation of Na v 1.5 [54]. This results in decreased surface membrane expression of SCN5A, leading to a reduction in the depolarizing current [53].

**SCN10A:** SCN10A encodes Na v 1.8, a neuronal sodium channel that appears to have a minor role in cardiac physiology. Prior genome-wide association studies (GWAS) have found an association between Na v 1.8 and BrS [55,56,57]. A recent study by Hu et al. identified SCN10A as a significant susceptibility gene associated with BrS. The study demonstrated that the co-expression of wild-type SCN5A (encoding the α-subunit of the cardiac sodium channel) with wild-type SCN10A leads to a gain of function in the sodium current (I Na), whereas the co-expression of wild-type SCN5A with mutant SCN10A results in a significant loss of function in iNa, leading to BrS. With the discovery of SCN10A as a susceptibility gene in approximately 16.7% of BrS probands, the identification of potentially causative mutations is now possible in over 50% of BrS patients. This finding enhances the utility of genotyping in risk stratification and screening of family members for BrS [50].

**SCN1B:** The SCN1B gene encodes the auxiliary Na V β1 subunit of the voltage-gated cardiac sodium channel. Following the initial descriptions by Watanabe et al., it was established that these mutations resulted in a loss of function of the peak sodium current [58]. Subsequent studies demonstrated that when mutant SCN1B was co-expressed with wild-type SCN5A (encoding the α-subunit of the sodium channel) and wild-type KCND3 (encoding the K V 4.3 potassium channel), it resulted in a significant 55.6% decrease in peak iNa and a 70.6% increase in Ito (transient outward potassium current) [59]. A co-immunoprecipitation analysis further revealed a structural association between Na V β1B, Na V 1.5, and K V 4.3, indicating that the elevated levels of the Ito play a critical role in these patients [59].

**SCN2B:** The SCN2B gene encodes the β2-subunit of NaV1.5. Mutations in this gene significantly decrease iNa density by reducing the expression of NaV1.5 on the cell surface and could thereby result in a BrS phenotype [60].

**SCN3B:** SCN3B encodes the β3 subunit of NaV1.5. Missense mutations in the SCN3B gene cause a reduction in the density of the peak sodium current, accelerating the inactivation process and slowing reactivation. These findings were reported by Hu et al., who also noted that these mutations impaired the intracellular transport and surface expression of cardiac sodium channels due to Navβ3 subunit dysfunction [61].

**PKP2:** The PKP2 gene encodes the desmosomal protein plakophilin-2 (PKP2), mutations of which have been identified in a small number of patients with features of BrS, without apparent structural cardiomyopathy [62]. These mutations can disrupt interactions between PKP2 and NaV1.5 at the cardiac intercalated disc, reducing the iNa (sodium current) and contributing to the development of BrS [63].

**FGF12:** FGF12, a newly discovered BrS-susceptibility gene, encodes fibroblast growth factor homologous factor 1, which modulates cardiac sodium and calcium channels. A Q7R missense mutation in FGF12, associated with BrS, has been found to decrease iNa [64,65].

**SLMAP:** SLMAP, a protein located in the sarcolemmal membrane and associated with T-tubules and sarcoplasmic reticulum, has been linked to BrS through genetic mutations. These mutations have been found to interfere with the trafficking of Na v 1.5, resulting in a loss of function in the sodium current (I Na) [66,67].

**RANGRF**: In a screening of patients with BrS and idiopathic VF, a missense mutation (p.Glu83Asp or E83D) was found in the RANGRF gene, which encodes MOG1. This study by Kattygnarath et al. [68] indicates that dominant-negative mutations in MOG1 can hinder the trafficking of Nav1.5 to the membrane, resulting in a reduction in iNa and the clinical presentation of BrS. In that study, silencing MOG1 also reduced the iNa density by 54%, suggesting its significance in the surface expression of Nav1.5 channels. Overall, these findings establish MOG1 as a novel susceptibility gene for BrS [68,69].

#### 2.3.1. Potassium Channels

Gain-of-function mutations in genes encoding potassium channels are identified in less than 1% of BrS patients. Genes KCNE3, KCND3, and SCN1B influence Ito [59,70,71], while KCNJ8 and ABCC9 mutations affect the IK-ATP channel [72,73,74].

**KCNE3 and KCND3:** BrS has been linked to the presence of gain-of-function mutations in KCNE3, also known as MiRP2 [70]. MiRP2 plays a crucial role in regulating various potassium currents in the heart, such as Ito and iKs. KCND3 encodes Kv4.3, the α-subunit of the Ito channel [71]. When KCNE3 mutations are co-expressed with wild-type KCND3, it results in an increase in the gain-of-function and accelerated kinetics of Ito [71].

**SCN1B:** In addition to reducing INa (sodium current), BrS-related mutations in the SCN1B gene, which encodes the auxiliary NaVβ1 subunit of the voltage-gated cardiac sodium channel, have been found to result in an enhancement of Ito (transient outward potassium current) when co-expressed with wild-type KCND3 [59].

**KCNJ8 and ABCC9:** Mutations in both KCNJ8 (Kir6.1) and ABCC9 (SUR2A) genes can result in a gain-of-function effect in the IK-ATP channel, leading to abnormal action potential characteristics associated with BrS or short QT syndrome phenotypes [72]. Gain-of-function mutations in the KCNJ8 gene, which encodes the Kir6.1 subunit of the ATP-sensitive potassium channel (IK-ATP), have been observed. This leads to the development of either BrS or short QT syndrome (SQTS) phenotypes by increased channel activity, resulting in an accentuated action potential notch and a depressed plateau phase [73]. Furthermore, recent studies have identified mutations in the ABCC9 gene as causative factors in BrS. This gene encodes the SUR2A subunit, an ATP-binding cassette transporter associated with the IK-ATP channel, and these mutations lead to a gain-of-function effect due to the diminished inhibitory effect of ATP [74].

KCNH2, which encodes for IKr, was also noted to be associated with BrS [55]. KCNAB2, which encodes the voltage-gated K+ channel β2-subunit, was most recently identified to be associated with increased Ito activity, contributing to the BrS phenotype [56].

#### 2.3.2. Calcium Channel Genes

L-type calcium channels (LTCCs) are responsible for calcium current conductance. There are four protein subunits of the LTCC-α1 (Cav1.2) subunit, which is a pore-forming unit that determines the main functional properties of the channel encoded by the CACNA1c gene, and three auxiliary subunits—the cytoplasmic β subunit, encoded by the CACNB; α2δ, encoded by CACNA2D; and a γ subunit, which is present in skeletal but not cardiac muscle. [57,58] Mutations in the α1-(CACNA1C), β2- (CACNB2), and α2δ- (CACNA2D1) are encountered in 2–4% of patients [59,60,61]. Loss-of-function mutations in these genes precipitate abnormal channel transport, resulting in reduced calcium influx during phase 2 [62]. Classically, BrS related to reduced calcium channels is associated with shorter QT intervals compared to classical SCN5A mutations [63].

Rare variants of these genes have additionally been described in short QT syndrome and idiopathic ventricular fibrillation [64,65]. Mutations in the L-type calcium channel genes are common with overlapping phenotypes of BrS with short QT syndrome but are rare in those without short QT intervals [46].

#### 2.3.3. Genes Affecting Ion Channels

**HCN4**: The HCN4 gene encodes potassium/sodium hyperpolarization-activated cyclic nucleotide-gated channel 4, which is mostly seen in the pacemaker region of mammalian hearts [66]. Mutations of this gene are associated with bradycardia and idiopathic VT [67].

**TRMP4**: The transient receptor potential melastatin protein 4 gene (TRPM4) encodes the calcium-activated nonselective ion channels responsible for the transport of monovalent cations across the plasmalemma and has been associated with BrS [68].

The variable penetrance of BrS is explained by a SCN5A polymorphism (H558R) on a discrete allele. Due to this, the mutation is fully rescued by the polymorphism, as the mutated channel can produce a functional channel before it traffics to the membrane. Hence, an assessment of the associated complementary alleles could prove pivotal in the risk stratification of these patients. This also identifies genetic polymorphisms as potential targets for novel therapies focusing on dysfunctional protein channels [69]. H558R has an allele frequency of around 10% in East Asian populations and approximately 20% in Caucasians [70]. This might explain the higher prevalence of manifest BrS in East Asians compared to Caucasians [69].

A recent study by Hosseini et al. assessed the clinical validity of genes tested for BrS using an evidence-based scoring system. The results revealed that out of the 21 genes evaluated, only 1 gene (SCN5A) demonstrated definitive evidence for disease causation, while the remaining 20 genes showed limited evidence. These findings underscore the importance of a systematic and evidence-based evaluation before utilizing these genes in patient care, highlighting concerns regarding the accuracy of reported gene–disease associations in BrS. Out of the 20 disputed genes, 19 were initially identified in candidate gene studies based on biological plausibility rather than unbiased genome-wide methods [75]. Among these, 13 genes (ABCC9, ANK2, CACNA2D1, HCN4, KCND3, KCNH2, KCNJ8, RANGRF, SCN1B, SCN2B, SCN3B, SLMAP, and TRPM4) were found to have rare variants only in sporadic cases, lacking segregation data or evidence of rare variant excess compared to control subjects. Three genes (CACNA1C, KCNE5, and SCN10A) showed limited genetic inheritance in two individuals across two generations, while only three genes CACNB2, KCNE3, and PKP2) had more than two instances of genetic inheritance within a family [63,70,76]. Similarly, while preconceived candidate gene studies implicated KCNE3 and PKP2 with four and three segregations, respectively, the statistical data, additional families, and supporting evidence in the literature were insufficient [63,70]. The early reports on gene–disease associations often relied on small control cohorts to determine variant rarity due to limited access to large databases. However, two curated genes (KCNJ8 and SCN3B), initially considered rare, were later found to have frequencies in public databases equal to or higher than the prevalence of the disease. This issue was further emphasized in a publication by Risgaard et al., which revealed that approximately 1 in 23 individuals carried a previously reported potential mutation associated with Brugada syndrome among ≈4000 publicly available exomes [77,78]. In conclusion, this study by Hosseini et al. emphasizes the higher standard required to establish a causal relationship between genes and diseases due to our growing understanding of natural genetic variations in the population. It highlighted the importance of cautious interpretation of functional assays that may not accurately reflect the disease phenotype or distinguish between rare benign variants [75].

### 2.4. Structural and Tissue Level Mechanisms

Initially, BrS was believed to have subtle structural abnormalities that were undetectable by standard diagnostic tools [29,79]. However, the discovery of SCN5A mutations in an ARVC family and recent evidence from CT scan studies suggest the presence of an abnormal structural RVOT component, including RV enlargement, abundant adipose tissue, and RV wall motion abnormalities correlating with the origin of spontaneous premature ventricular contractions (PVCs) after an arrhythmic event [80,81]. Additionally, an examination of the heart from a BrS patient with an SCN5A mutation and electrical storms showed substantial structural derangements, such as fatty replacement and intense fibrosis, specifically in the RVOT, while the left ventricle remained normal [82]. Recent studies have not only confirmed this association between structural and functional abnormalities but have also hypothesized that the functional disturbances, particularly the reduction in sodium current (I_Na_), might be responsible for initiating the observed structural abnormalities [83,84]. Another possible explanation for the conduction slowing in the RVOT in BrS involves the presence of specialized tissues similar to cardiac nodes, which rely on an L-type calcium current (I_Ca-L_) for their action potential upstroke. It is hypothesized that remnants of these node-like cells in the RVOT could serve as the underlying substrate for arrhythmias that originate in this region [85,86]. There was a debate on whether BrS was in fact a form of RV dysplasia (ARVC). It is hypothesized that molecules such as desmosomes, gap junctions, and sodium channel complexes interact within a connexome network to control electrical coupling and intercellular adhesion in the heart, suggesting a potential common origin for ARVC and BrS [87]. While there may be structural abnormalities, follow-up of a large amount of patients has not shown that there is a progressive structural heart disease. Minor structural abnormalities in the right ventricle have been observed in some BrS patients, and mutations in the SCN5A gene have been linked to a dilated cardiomyopathy [88]. Similarly, mutations in desmosomal genes can lead to ventricular fibrillation and sudden death, even in the absence of apparent structural heart disease [89,90]. Cerrone et al. conducted a study exploring whether mutations in the PKP2 gene, which play a role in the desmosomal function, could be present in individuals with BrS. The authors examined the PKP2 gene in two-hundred BrS patients who did not have mutations in the commonly implicated genes and identified five distinct mutations in five unrelated patients. Subsequent experiments using cell lines and human cardiomyocytes were performed to investigate the relationship between PKP2 mutations and the electrical activity of the heart. These findings provide the initial evidence that missense mutations in the PKP2 gene can reduce INa and facilitate arrhythmias, even in the absence of structural heart disease [63,91]. Consequently, although these conditions are clinically different, they may share a common origin as disorders related to the connexome.

In another interesting hypothesis aimed at providing a unified explanation, Elizari et al. suggested that the abnormal expression of neural crest cells during the embryological development of the RVOT and its surrounding structures may be linked to Brugada syndrome (BrS). This abnormal expression could lead to delayed depolarization and repolarization heterogeneity in the RVOT, resulting in the observed ECG characteristics of Brugada syndrome [92]. The distribution of connexin molecules, particularly Cx43, which play a role in neural crest cell migration and cardiac impulse propagation, may be crucial in the development of Brugada syndrome [93]. In a study involving six postmortem cases, it was found that patients with Brugada syndrome had increased fibrosis in the heart, specifically in the RVOT and epicardium, along with reduced expression of Cx43. The presence of fibrosis correlated with decreased gap junction expression and abnormal action potentials. However, the phenotype of BrS and associated ventricular arrhythmias were eliminated through ablation procedures [94].

## 3. Clinical Features and Management

The majority of individuals with BrS do not exhibit clinical symptoms and are diagnosed based on specific EKG findings, termed the BrP. Currently, the diagnosis of BrS involves identifying a characteristic pattern of ST-segment elevation, specifically referred to as coved-type or type 1 morphology, with a measurement of ≥2 mm in ≥1 leads from V1 to V2, followed by a negative T wave. This may occur spontaneously, be provoked by certain drugs (sodium channel blockers), or be induced by fever and exercise tests [95]. Vitali et al. conducted a review on BrS patients and identified 12 ECG signs that serve as high-risk markers, including specific localization of the type 1 Brugada pattern, first-degree atrioventricular block, atrial fibrillation, fragmented QRS, prolonged QRS duration, elevated R wave in lead aVR, the presence of S wave in L1 meeting specific criteria, ER patterns in inferolateral leads, ST-segment depression, T-wave alternans, the dispersion of repolarization, and adherence to the Tzou criteria [96]. Type 1 is considered diagnostic of the syndrome, while type 2 and type 3 patterns are not diagnostic, but may still be observed in patients. Type 2 has a saddleback appearance with an ST-segment elevation of ≥2 mm, a trough with ≥1 mm ST elevation, and a positive or biphasic T wave. Type 3 can have either a saddleback or coved appearance, with an ST-segment elevation of <1 mm. It is important to note that type 2 and type 3 ECG patterns alone are not sufficient for diagnosing Brugada syndrome. However, if a patient exhibits a type 2 or type 3 ST-segment elevation in more than one right precordial lead under baseline conditions and there is a conversion to the diagnostic type 1 pattern after the administration of a sodium channel blocker, then the diagnosis of Brugada syndrome is considered positive [10]. The severity of symptoms can vary, ranging from syncope (fainting) and nocturnal agonal breathing to ventricular arrhythmias and sudden cardiac death (SCD). These symptoms are observed in approximately 17% to 42% of BrS patients, with an initial presentation of SCD in nearly 5% of cases [97]. The expansion in the identification of asymptomatic individuals has contributed to the overall dilution in event rates.

BrS commonly presents during the third or fourth decade of life, with an average age of onset around 41 years. The presence of a Brugada type 1 ECG pattern is associated with an increased risk of cardiac events per year in patients with aborted SCD (7.7%) or symptoms of syncope (1.9%) compared to asymptomatic individuals (0.5%) [98]. Approximately 20% of individuals may experience arrythmias such as atrial flutter, atrial fibrillation (AF), AV nodal reentry, and pre-excitation syndromes such as Wolff–Parkinson–White (WPW) syndrome [99]. Ventricular arrhythmias in BrS tend to occur at rest, particularly during nighttime or sleep, with a study by Matsuo et al. indicating that 26 out of 30 documented episodes of VF in BrS patients with an implantable cardioverter defibrillator (ICD) occurred during sleep, suggesting a potential involvement of vagal activity in the arrhythmogenesis of the syndrome [100]. The nocturnal association of arrhythmic events in BrS has been attributed to elevated vagotonic influences and sleep-disordered breathing, which are considered high-risk characteristics observed in up to 63% of patients with characteristic ST-T changes during REM sleep and arousals [101].

The clinical presentation of BrS is significantly more common in males, with a prevalence 8 to 10 times higher than in females [102]. As a result, previous studies have predominantly included a male population of 71% to 77% and have shown that males with BrS exhibit a higher risk profile compared to females, displaying more frequent symptoms at the time of diagnosis [47,48,103,104]. Two main hypotheses have been proposed to explain these sex differences. One theory by Diego et al. suggests that the differences may be due to intrinsic disparities in ionic currents between males and females. The density of Ito is higher in male right ventricle epicardia compared to females, which contributes to the deeper notch in phase 1 of the AP in males. This difference in phase 1 magnitude can lead to ST-segment elevation and increase the risk of phase 2 reentry and ventricular fibrillation in males [105]. The other hypothesis suggests that sex hormones might also play a role in the phenotypic manifestations of Brugada syndrome. Some evidence suggests that testosterone levels are higher in male Brugada patients compared to controls, and the regression of ECG features has been observed in castrated men [106,107,108]. However, the exact mechanism by which hormones affect the ionic membrane currents in humans is still unclear, and the role of sex hormones in BrS among children is not yet established.

Limited data are available on the behavior of BrS in childhood, but studies indicate that the syndrome can manifest during childhood, with symptoms potentially appearing during febrile episodes [109]. Symptomatic patients, especially those with a spontaneous type 1 ECG, may be at a higher risk of cardiac events.

Several conditions and medications have been identified as potential triggers for arrhythmic events and sudden cardiac death in BrS, as they can exacerbate the underlying electrophysiological mechanisms involved in the disorder. These mechanisms include enhanced sodium blockade, augmentation of I_to_, phase 2 calcium and sodium channel activation, and increased vagal tone. Fever is recognized as a significant trigger for cardiac arrest in individuals with this disorder, affecting approximately 18% of patients [110]. In three cases of fever-induced cardiac arrest, patients were identified as having an SCN5A mutation, which impairs cardiac sodium channel function in higher temperatures [111]. Additionally, familial diseases characterized by temperature-dependent symptoms, including generalized epilepsy with febrile seizures (associated with SCN1A mutations) and inherited erythromelalgia with heat-triggered pain and skin redness (associated with SCN9A mutations), have been linked to mutations in genes encoding neuronal sodium channels [112,113]. As a result, aggressive antipyretics are considered crucial in the management of these patients. Interestingly, studies have indicated that Asians have a lower occurrence of fever-related arrhythmic events compared to Caucasians, with rates of 2.9% and 8.5%, respectively [114]. Excessive alcohol consumption has also been found to be associated with higher lethal arrhythmic events in Japanese patients with BrS. This is particularly relevant due to the presence of loss-of-function variants of alcohol-metabolizing enzymes, such as ADH1B and ALDH2, which are more prevalent in this population [115,116]. The ADH1B His/His variant, which is present in about 60% of the Japanese population, exhibits significantly greater alcohol-metabolizing activity than the ADH1B Arg/Arg variant, leading to elevated blood acetaldehyde levels and increased sympathetic nervous system stimulation, which can trigger arrhythmic events [117,118].

The dynamic nature of the ECG can sometimes hide the characteristic patterns of BrS, proposing the use of sodium channel blockers for diagnostic purposes [119]. While drugs like ajmaline, flecainide, procainamide, pilsicainide, disopyramide, and propafenone have been utilized, their individual diagnostic value has not been extensively investigated. The accuracy of pharmacological provocation in terms of sensitivity and specificity is still debated due to the lack of comprehensive studies involving sequential drug testing with different agents in patients [10,120]. The current evidence suggests that ajmaline is likely the most effective drug for revealing BrS. A study involving 147 individuals with known SCN5A mutations showed that the use of ajmaline (sensitivity of 80%; specificity of 94.4%) increased the penetrance of the disease phenotype from 32.7% to 78.6%, surpassing the diagnostic performance of flecainide (sensitivity of 77%; specificity of 80%) in a study with 110 genotyped patients [112,121]. Recent studies have also examined clinical risk score models to predict malignant arrhythmic events in patients with BrS. The Shanghai Brugada scoring system showed predictive value for arrhythmic events in asymptomatic patients, those with syncopal episodes, and those with previous ventricular fibrillation [122]. Another simplified risk model was developed by Sieira et al. based on six items, including novel markers like sinus node dysfunction and early family history of sudden cardiac death that could effectively predict the likelihood of arrhythmic events in BS patients and assist in their management. The model exhibited strong predictive performance in the training cohort and maintained its effectiveness when tested on a distinct cohort of 150 patients, even though their risk profiles varied. While determining a threshold for high-risk classification remains difficult, an estimated 5-year event risk of 6% (equivalent to a score of two points in the model) could be considered, with individualized decision-making based on objective data, patient preferences, and clinician expertise [123]. In another multicentric study, Probst et al. conducted a study with 1613 BrS patients to evaluate the Shanghai score and Sieira score for risk assessments. The findings showed that while both scoring systems were effective in identifying arrhythmic event risks in patients with high or low scores, it was challenging to stratify the risk for patients with intermediate scores [124].

Currently, the only proven effective treatment for BrS is the implantable cardioverter-defibrillator (ICD), according to a 2003 consensus conference. Electrophysiological studies (EPS) can be performed in symptomatic patients to assess the test’s sensitivity and specificity for outcome prediction, as well as to study supraventricular arrhythmias. In asymptomatic patients, EPS can be beneficial for risk stratification, with ICD implantation recommended for those who have inducible ventricular fibrillation (VF) and specific EKG patterns, especially if there is a positive family history of sudden death. However, for asymptomatic patients without a family history, but who only exhibit a type 1 ECG after sodium channel blockade, close monitoring is advised, as there is limited evidence supporting the use of EPS or a direct indication for ICD in such cases [10]. Instead, pharmacologic approaches have been explored to rebalance currents during the early phases of the epicardial action potential. While drugs like amiodarone and β blockers have proven ineffective, quinidine has shown promise in normalizing ST-segment elevation and preventing arrhythmic events. Quinidine may serve as an adjunct or alternative to ICD therapy, and its potential benefits in asymptomatic patients require further evaluation [125]. In a preliminary study by Hermida et al., it was found that hydroquinidine (HQ) therapy showed promising results in preventing VT and VF in asymptomatic patients with BrS. In a patient with multiple ICD shocks, HQ successfully prevented further shocks and episodes of VT/VF [126]. Prior studies have also reported the effectiveness of quinidine in preventing VT/VF inducibility and the recurrence of arrhythmic events in patients with BrS [114]. Monitoring the QTc interval is crucial in assessing the cardiac tolerance and compliance of HQ therapy, as major QTc prolongation and syncope have been observed in some case. Developing more specific blockers and investigating agents that augment L-type calcium channel (isoproterenol and cilostazol) currently offer possibilities for enhancing treatment options. Additionally, the experimental agent tedisamil, which blocks Ito, has been proposed as a potential therapeutic option [105]. In the long-awaited results of the BRAVO study (Brugada Ablation of VF Substrate Ongoing), positive outcomes in the management of BrS through RFA were observed. Patients with high-risk characteristics and documented ventricular fibrillation experienced a significant reduction in arrhythmic episodes following epicardial ablation [127]. Further studies, such as BRAVE (Ablation in Brugada syndrome for the Prevention of VF), should explore the potential benefits of combining implantable cardioverter–defibrillator (ICD) therapy with ablation, and consider the use of ablation in asymptomatic patients with BrS cautiously, awaiting more robust evidence [128].

## 4. Clinical Significance of Genetic Testing

Currently, molecular genetic testing can only detect mutations in a limited subset of BrS patients, ranging from 20% to 38%. Therefore, routine screening of asymptomatic individuals is not considered justified at this time [47,129]. However, there are several reasons that support genetic testing for patients with a confirmed diagnosis of BS, such as the high prevalence of affected parents among BrS patients, the rarity of cases caused by spontaneous mutations (approximately 1%), the significant likelihood of disease manifestation (around 30% of individuals with an SCN5A mutation exhibit ECG changes characteristic of BS, which increases to approximately 80% with the use of sodium channel blockers), and the limited effectiveness of ECG changes in diagnosing at-risk relatives [130]. When a disease-causing mutation has been previously identified in the family, it may be more appropriate to consider predictive testing for relatives (Figure 2). Molecular genetic testing can help identify individuals at a higher risk of adverse outcomes, enabling preventive measures such as closer monitoring and avoiding triggers for ventricular arrhythmias. It is important to note that in cases where the mutation cannot be detected in the parents, other possibilities such as germline mosaicism, de novo mutations, alternate paternity or maternity, or undisclosed adoption should be considered. For affected patients, preimplantation genetic diagnosis and prenatal testing are viable options for at-risk pregnancies, but it is advisable to first identify the disease-causing mutation within the family [131]. Currently, there is no reliable laboratory test with sufficient diagnostic or prognostic value for BrS, and the usefulness of biochemical markers of cardiac injury is uncertain. While some cases have shown increased levels of cardio-specific troponins in patients with BS, other studies have reported negative findings, suggesting that elevated biomarkers may indicate complications like acute myocardial infarction rather than a direct pathophysiological process [132,133].

## 5. Human-Induced Pluripotent Stem Cell-Derived Cardiomyocytes and Model for Precision Medicine

Human-induced pluripotent stem cell-derived cardiomyocytes (hiPSC-CMs) offer a valuable model for studying BrS and advancing precision medicine. These cells, generated through reprogramming adult cells, allow for the ethical study of cardiovascular disease mechanisms [134]. Recent advancements in differentiation protocols have enabled the generation of atrial-specific hiPSC-CMs, facilitating the investigation of chamber-specific remodeling and BrS. Researchers can direct hiPSC-CMs into atrial-like or ventricular-like phenotypes by modulating signaling pathways [135,136]. Other studies have successfully directed hiPSC-CMs toward sinoatrial node-like cells through the combined modulation of BMP, fibroblast growth factor, and retinoic acid signaling pathways [137,138]. In a study conducted by Davis et al., hiPSC-CMs derived from heterozygous mice were utilized to investigate the electrophysiological properties of sodium channel mutations. The results demonstrated the suitability of hiPSC-CMs as models for cardiac sodium channel diseases [139]. Another study by Cerrone et al. revealed that missense mutations in plakophilin-2 (PKP2) could lead to sodium current deficits, specifically decreased INa and Nav1.5 at the site of cell contact. The deficit was shown to be restored by transfection of the wild-type PKP2, indicating that PKP2 mutations could serve as a molecular substrate for BrS [63]. In a prior study by Antzelevitch et al., it was demonstrated that introducing gene mutations into Chinese hamster ovary K1 cells resulted in a decrease in the calcium current [76]. In another study by Miller et al., it was revealed that hiPSC-CMs obtained from individuals with Brugada syndrome (BrS) without known pathogenic mutations exhibited comparable action potential parameters to control hiPSC-CMs [140].

The SCN5A gene, associated with the sodium channel, has been extensively studied in BrS. HiPSC-CMs derived from patients with SCN5A mutations exhibit reduced sodium channel functions, altered action potentials, abnormal calcium handling, and conduction defects [62]. Other genes linked to sodium, calcium, and potassium channels have also been implicated in BrS, but their effects on hiPSC-CMs remain largely unexplored. To investigate disease mechanisms and potential treatments, researchers are utilizing CRISPR/Cas9-mediated genome editing in combination with hiPSC techniques. This approach allows for the modification of disease-causing genes and the study of pathophysiology, drug screening, and cell therapy enhancement in BrS [141].

While previous studies focused on single-cell phenotypes, researchers are now turning the integration of 3-dimensional and 2-dimensional models using hiPSC-CMs, which holds significant potential for advancing preclinical drug screening and supporting the development of cardiovascular tissue engineering in the context of precision medicine [142]. Zimmermann et al. conducted a study demonstrating that 3D-engineered heart tissue derived from rat cardiac myocytes exhibited improved tissue formation and prevented excessive growth of noncardiomyocytes, indicating its retention of key physiological characteristics. Additionally, the study highlighted the efficient gene transfer and subsequent force measurement capabilities of this engineered tissue, suggesting its potential applications in drug discovery [143]. The use of hiPSC-CMs in BrS research contributes to a better understanding of disease mechanisms, the identification of potential drug targets, and the testing of drug toxicity. Studying the pathogenicity of gene variants and conducting drug screening using hiPSC-CMs aid in improving risk stratification and developing personalized treatments for BrS patients. With the combination of emerging technologies like genome editing and tissue engineering, hiPSC-CMs are poised to play an increasingly important role in BrS research.

## 6. Conclusions

The current understanding of the genetic and molecular mechanisms underlying BrS is limited. The complexity of this condition poses challenges in every domain of care—diagnosis, risk assessment, and management. However, ongoing research is shedding light on the genetic and molecular aspects, including the increasing recognition of the role of structural abnormalities. Going forward, the integration of these findings into clinical decision-making and prognostication could play a pivotal role in the management of these patients. Developing and validating a comprehensive risk score that incorporates clinical, genetic, electrocardiographic, electrophysiologic, and environmental factors could improve the prediction of arrhythmia and sudden cardiac death, leading to better therapeutic management.

## Figures and Tables

**Figure 1 cells-12-01791-f001:**
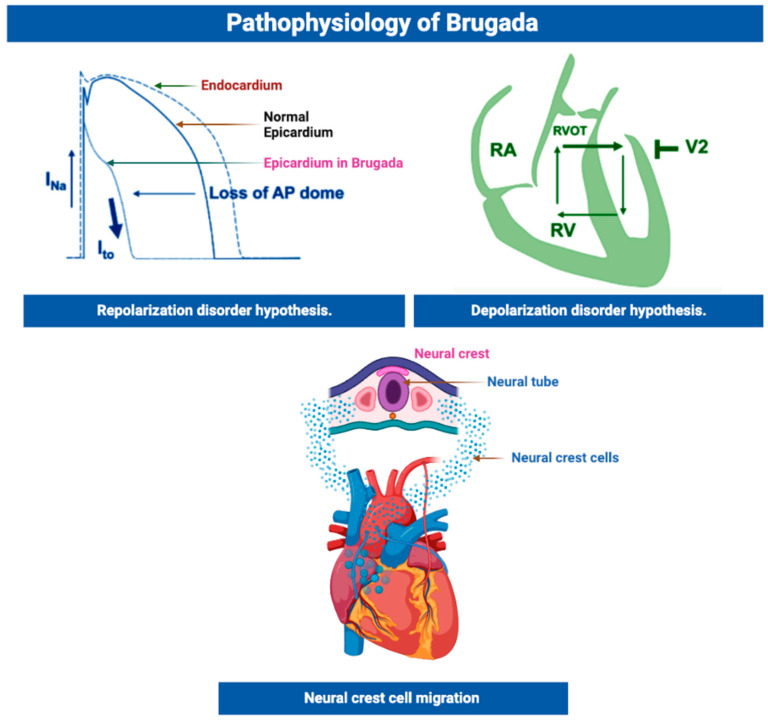
Overview of pathophysiology.

**Figure 2 cells-12-01791-f002:**
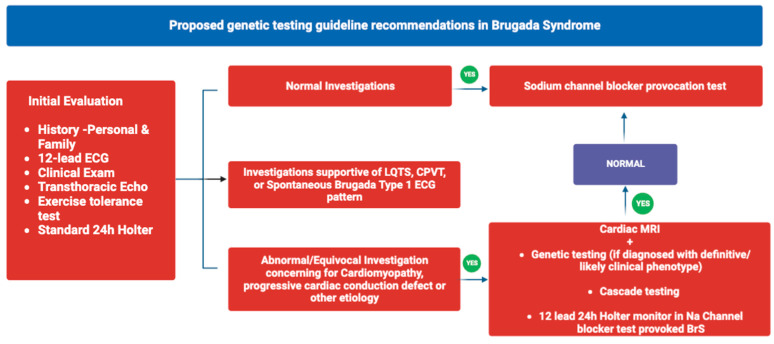
Overview of genetic testing guideline recommendations.

## Data Availability

Not applicable.

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
