# Peer review of "Genetic and Molecular Mechanisms in Brugada Syndrome"

_cells, 2023, doi:10.3390/cells12131791_

Round 1

Reviewer 1 Report

This manuscript is a well-written and concise review of BrS and the authors have done a good job of summarizing a huge body of evidence to a few pages.

One minor remark:

As the authors reported the use of Quinide in the therapy of BrS one would suggest elaborating on the pharmacodynamic effect of the drug and its supposed mechanism for treating BrS despite being a Class I Na channel blocker, i.e. the Ito blocking properties of the drug. This could also be noted in the mechanistic biochemical part of the manuscript as one more evidence of Ito relevance in Brs.

Author Response

Reviewer #1

Comment: This manuscript is a well-written and concise review of BrS and the authors have done a good job of summarizing a huge body of evidence to a few pages.

One minor remark:

As the authors reported the use of Quinide in the therapy of BrS one would suggest elaborating on the pharmacodynamic effect of the drug and its supposed mechanism for treating BrS despite being a Class I Na channel blocker, i.e. the Ito blocking properties of the drug. This could also be noted in the mechanistic biochemical part of the manuscript as one more evidence of Ito relevance in Brs

Response: Page 13, Line 575-582, we sincerely appreciate your valuable feedback and constructive suggestion. We have elaborated on the pharmacodynamic effect of the drug and its supposed mechanism for treating BrS

Reviewer 2 Report

This review focuses on the pathophysiology, genetics and molecular mechanisms underlying Brugada syndrome (BrS), a cardiac channelopathy associated with an increased risk of sudden cardiac death. It is well-written, well-structured and informative. I only have minor suggestions.

The pathophysiology section would benefit from a paragraph explaining the contributions of the different currents towards generation of an action potential. This would facilitate understanding of changes underlying the repolarization and depolarization theories.

Lines 141-142; I suggest the authors refer to the overlap between BrS and ARVC further down in the manuscript after they have covered the genetics. In the overlap section, in addition to genetics, fibrosis and gap junction remodelling, they should also refer to the inflammatory infiltrates found in both BrS and ARVC myocardium.

The section on the connexome (lines 138-140) would benefit from more details as well as addition of a diagram. Similar to above, it should be moved further down in the manuscript.  

In the section on SCN5A, I suggest addition of a sentence stating that SCN5A mutations are also found to underlie other conditions apart from BrS.

Is there a current theory explaining the greatly increased incidence of BrS in males compared to females?

ICD therapy lines 432-433; Perhaps emit the sentence as higher up the authors have stated that BrS is manifested in childhood in very rare cases and there is no report of manifestation in infancy.

I suggest adding more details on the two types of ECG patterns.

Author Response

Reviewer #2:

Comment: This review focuses on the pathophysiology, genetics and molecular mechanisms underlying Brugada syndrome (BrS), a cardiac channelopathy associated with an increased risk of sudden cardiac death. It is well-written, well-structured and informative. I only have minor suggestions.

The pathophysiology section would benefit from a paragraph explaining the contributions of the different currents towards generation of an action potential. This would facilitate understanding of changes underlying the repolarization and depolarization theories.
Response: Page 3, Lines 92 – 105, we have added a section explaining the role of different currents towards generation of an action potential.

Comment: Lines 141-142; I suggest the authors refer to the overlap between BrS and ARVC further down in the manuscript after they have covered the genetics. In the overlap section, in addition to genetics, fibrosis and gap junction remodelling, they should also refer to the inflammatory infiltrates found in both BrS and ARVC myocardium.
Response: Page 10, Lines 412 – 430, moved it further down. We did not find any strong data which correlates the inflammatory infiltrates found in both BrS and ARVC myocardium. However, we did elaborate a little on the common origin for ARVC and BrS in this section

Comment: The section on the connexome (lines 138-140) would benefit from more details as well as addition of a diagram. Similar to above, it should be moved further down in the manuscript.  
Response: Page 10, Lines 412 – 444, Moved it further down and expanded on the distribution of connexin molecules, its role in the common origin for ARVC and BrS, and its role in neural crest cell migration.

Comment: In the section on SCN5A, I suggest addition of a sentence stating that SCN5A mutations are also found to underlie other conditions apart from BrS.
Response: Page 7, Lines 257 – 264, expanded on other conditions due to SCN5A mutations

Comment: Is there a current theory explaining the greatly increased incidence of BrS in males compared to females?
Response: Page 11, Lines 492 -503, elaborates on the 2 main hypothesis that could potentially explain the increased incidence of BrS in males compared to females

Comment: ICD therapy lines 432-433; Perhaps emit the sentence as higher up the authors have stated that BrS is manifested in childhood in very rare cases and there is no report of manifestation in infancy.
Response: Agree, sentence has been removed

Comment: I suggest adding more details on the two types of ECG patterns
Response: Page 11, Lines 459 468, expanded on the two types of ECG patterns seen in BrS

Reviewer 3 Report

Moras et al. present in a review genetics and molecular pathomechanisms of brugada syndrome. Brugada syndrome is an inherited channelopathy. Patients are at high risk of sudden cardiac death at young age. Brugada syndrome has been described 1992 and since their a relevant progress including genetic findings have been achieved. The review article is well written. I have the following suggestions

1-Abstract should give more details (hypothesis of BrS, role of sodium channel current etc). Of note the rate oof 30-35% seems to be high. In addition, recent statement paper reevaluated the published genes and only SCN5A has been linked to BrS. PMID: 29959160

2-Genetic mechanisms:

Please discuss the recent reappraisal of BrS gene PMID: 29959160.

The hypothesis that BrS is polygenic disease needs to be discussed PMID: 35322667

What is the role of induced pluripotent stem cell models. There are a couple focusing on the described BrS genes and they need to be integrated.

3-Please discuss the use of current models e.g. induced pluripotent models for precision medicine and personalized therapy.

What about drug treatment in BrS?

4-Relevant references for understanding the pathophysiology in BrS are missing. This needs to be revised.

5-What is the link between BrS and fever.

Author Response

Reviewer #3

Comment: Moras et al. present in a review genetics and molecular pathomechanisms of brugada syndrome. Brugada syndrome is an inherited channelopathy. Patients are at high risk of sudden cardiac death at young age. Brugada syndrome has been described 1992 and since their a relevant progress including genetic findings have been achieved. The review article is well written. I have the following suggestions

1-Abstract should give more details (hypothesis of BrS, role of sodium channel current etc). Of note the rate oof 30-35% seems to be high. In addition, recent statement paper reevaluated the published genes and only SCN5A has been linked to BrS. PMID: 29959160
Response: Page 1, Line 19-33, We have completely revised the abstract to give more details on the hypothesis of BrS, role of sodium channel current etc. In addition, we have taken into consideration the recent paper PMID: 29959160 and added relevant sections in the abstract with more details in the manuscript.

Comment: 2-Genetic mechanisms: Please discuss the recent reappraisal of BrS gene PMID: 29959160.
Response: Page 9-10, Lines 386 – 393, we have integrated this recent reappraisal of BrS gene into this section.

Comment: The hypothesis that BrS is polygenic disease needs to be discussed PMID: 35322667
What is the role of induced pluripotent stem cell models. There are a couple focusing on the described BrS genes and they need to be integrated.
Please discuss the use of current models e.g., induced pluripotent models for precision medicine and personalized therapy.
Response: Page 14, Lines 627 – 654, we reviewed this study PMID: 35322667 by Li et al and have dedicated a section at the end of our review addressing the hypothesis that BrS is polygenic disease, the role of induced pluripotent stem cell models and the use of current models such as induced pluripotent models for precision medicine and personalized therapy.

Comment: What about drug treatment in BrS?
Response: Page 13, Lines 569-585. Elaborates on drug treatment for BrS including novel therapies. In addition, we have expanded on the role of Quinidine therapy, its pharmacodynamic effect and its supposed mechanism for treating BrS as requested by Reviewer #1

Comment: 4-Relevant references for understanding the pathophysiology in BrS are missing. This needs to be revised.
Response: Page 2 – Page 10 , which elaborate on different pathophysiological mechanisms in BrS have been updated on references that we found relevant to the section.

Comment: 5-What is the link between BrS and fever.
Response: Page 12, Lines 514 – 520, expands further on the relationship between fever and BrS

Round 2

Reviewer 3 Report

Thank you for revising the paper. Relevant papers are still missing. I recommend strongly to provide them PMID: 29959160 and to discuss this issuse.

In the section induced pluripotent stem cells no papers about relevant BrS models using this technology are provided. This source, PMID: 35322667, may assist you to provide relevant papers and discuss the use of them for understanding insights into BrS. 

Author Response

                                                                                               NYU School of Medicine          

                                                                                               550 First Avenue, New York, NY 10016              

                                                                                               Chayakrit Krittanawong, M.D.

June 30, 2023 

Editor

Cells

Subject: Rebuttal Letter for Manuscript ID #2480817

Dear the Editor

We are pleased to resubmit the invited manuscript cells-2480817 entitled, “Genetic and molecular mechanisms in Brugada Syndrome”.

We have carefully reviewed the Editors’ comments and have revised the manuscript point by point and resubmitted two versions of revised documents (with the track changes ON and clean version).

The manuscript, as submitted or its essence in another version, is not under consideration for publication elsewhere and will not be published elsewhere either in whole or in part, nor have the findings been posted online while under consideration by the Cells. The author has no commercial associations or sources of support that might pose a conflict of interest.

We hope these modifications will be met with your approval. Thank you very much.

Sincerely yours,

Chayakrit Krittanawong, MD

Reviewer comments

Reviewer #3

Comment: Thank you for revising the paper. Relevant papers are still missing. I recommend strongly to provide them PMID: 29959160 and to discuss this issuse.

Response: Page 10, Lines 330 – 352, integrated relevant papers from PMID: 29959160  

In the section induced pluripotent stem cells no papers about relevant BrS models using this technology are provided. This source, PMID: 35322667, may assist you to provide relevant papers and discuss the use of them for understanding insights into BrS. 
Response: Page 14 -15, Lines 588 – 602, 616-621, integrated relevant papers discussing about relevant BrS models using this technology from PMID: 35322667
